# Predictors of Sexual Dysfunction in Veterans with Post-Traumatic Stress Disorder

**DOI:** 10.3390/jcm8040432

**Published:** 2019-03-29

**Authors:** Marina Letica-Crepulja, Aleksandra Stevanović, Marina Protuđer, Božidar Popović, Darija Salopek-Žiha, Snježana Vondraček

**Affiliations:** 1Department of Psychiatry and Psychological Medicine, Faculty of Medicine, University of Rijeka, 51000 Rijeka, Croatia; aleksandras@medri.uniri.hr; 2Department of Psychiatry, Clinical Hospital Center Rijeka, Referral Center of the Ministry of Health of the Republic of Croatia, 51000 Rijeka, Croatia; 3Department of Basic Medical Sciences, Faculty of Health Studies, University of rijeka, 51000 Rijeka, Croatia; 4County General Hospital Varaždin, 42000 Varaždin, Croatia; marina2ri@yahoo.com; 5County General Hospital Našice, 31500 Našice, Croatia; salutogeneza1@gmail.com (B.P.); mentordsz@gmail.com (D.S.-Ž.); bolnica@obnasice.hr (S.V.)

**Keywords:** post-traumatic stress disorder, sexual dysfunction, veterans, predictors

## Abstract

Background: The problems in sexual functioning among patients with post-traumatic stress disorder (PTSD) are often overlooked, although scientific research confirms high rates of sexual dysfunctions (SD) particularly among veterans with PTSD. The main objective of this study was to systematically identify predictors of SD among veterans with PTSD. Methods: Three hundred veterans with PTSD were included in the cross-sectional study. The subjects were assessed by the Mini-International Neuropsychiatric Interview (MINI) and self-report questionnaires: PCL-5, i.e., PTSD Checklist for Diagnostic and Statistical Manual of Mental Disorders, Fifth Edition (DSM-5) with Criterion A, International Index of Erectile Function (IIEF), Premature Ejaculation Diagnostic Tool (PEDT), and Relationship Assessment Scale (RAS). Several hierarchical multiple regressions were performed to test for the best prediction models for outcome variables of different types of SD. Results: 65% of participants received a provisional diagnosis of SD. All tested prediction models showed a good model fit. The significant individual predictors were cluster D (Trauma-Related Negative Alterations in Cognition and Mood) symptoms (for all types of SD) and in a relationship status/relationship satisfaction (all, except for premature ejaculation (PE)). Conclusions: The most salient implication of this study is the importance of sexual health assessment in veterans with PTSD. Therapeutic interventions should be focused on D symptoms and intended to improve relationship functioning with the aim to lessen the rates of SD. Psychotropic treatment with fewer adverse sexual effects is of utmost importance if pharmacotherapy is applied. Appropriate prevention, screening, and treatment of medical conditions could improve sexual functioning in veterans with PTSD.

## 1. Introduction

The problems in sexual functioning among patients with post-traumatic stress disorder (PTSD) are often overlooked clinically and receive little attention in research. However, an increasing body of scientific research regarding sexual dysfunctions (SD) among veterans who were exposed to military trauma confirms much higher rates of problems in sexual functioning among veterans with PTSD than in those without PTSD or in adults without exposure to military trauma [1,2,3,4,5]. The rates of SD differ across the studies, mainly because of methodological differences. Systematic reviews reported a prevalence of SD between 8.4% and 88.6% among male veterans with PTSD [3,5]. Persons with PTSD, compared with similarly exposed survivors without it, have an increased risk of SD implying that PTSD, rather than trauma exposure per se, is the more proximal antecedent to sexual problems [3,6,7,8,9,10]. Studies revealed correlation of PTSD with a variety of impairments in the specific domains of sexuality (desire, arousal, orgasm, resolution) [1,2,3,4,5,6,7]. On the other hand, the specific PTSD symptoms or PTSD symptom clusters may influence the prevalence of SD unevenly. The emotional numbing and avoidance cluster, for example, appeared to be intimately tied to impairment in sexual functioning and higher level of sexual anxiety [2,11,12]. 

### 1.1. Predictors of Sexual Dysfunction in Veterans with PTSD

Only a few studies and systematic reviews have addressed the possible predictors that have an impact on sexual functioning in the population of veterans with or without PTSD. Considering the relationship between overall PTSD symptom severity and SD, studies revealed conflicting results [5]. Particular PTSD clusters and symptoms have been studied, and it was hypothesized that autonomic arousal, anger/hostility [13], emotional numbing/avoidance symptoms [2,11,12], and chronic autonomic arousal and intrusive symptoms [3,14,15] were mostly associated with sexual problems among veterans with PTSD. Recent studies indicate that emotional numbing may impede intimacy and attachment, thus serving as a potential mechanism through which symptoms of PTSD may drive problems and predict SD in these patients. According to the Diagnostic and Statistical Manual of Mental Disorders, Fifth Edition (DSM-5) [16], numbing symptoms (low positive emotions and negative emotional state) were included in the new D symptom cluster (Trauma-Related Negative Alterations in Cognition and Mood). These and other symptoms from this cluster, such as diminished interest or participation in significant activities, a feeling of detachment or estrangement from others, and guilt and shame, may impede sexual functioning in veterans with PTSD. SD is more common among veterans who are male, older, separated, divorced, or widowed, have lower annual income, mental health diagnoses—particularly PTSD—hypertension, and are prescribed psychiatric medications [1,4,17]. Returning combat veterans with SD have a reduced quality of life, decreased sexual intimacy, and increased health-care utilization [18]. PTSD is associated with impairments in romantic relationship satisfaction [19,20]. Recent research revealed that marital dissatisfaction is the factor that mediates the relationship between the number of PTSD symptoms and sexual dissatisfaction [21]. Considering the specific types of SD, age appeared to be the only significant predictor of erectile dysfunction; age, race, depression, and social support predicted self-reported sexual arousal problems; and race, combat exposure, social support, and avoidance/numbing symptoms of PTSD predicted self-reported sexual desire problems in male combat veterans seeking outpatient treatment for PTSD [2]. 

### 1.2. Predictors of Sexual Dysfunction in the General Population

Generally speaking, the predictors, risk, or etiological factors of SD can be separated in two groups: “organic” (such as diabetes, peripheral vascular disease or venous leaks, injury of the spinal cord, etc.) and “non-organic” (such as anxiety, depression, cultural taboos, ignorance, relationship problems, poor communication skills, etc.). However, there is substantial evidence indicating a multifactorial etiology of sexual function and dysfunction, meaning that the sexual response can be described as a complex interaction of psychological, interpersonal, social, cultural, physiological, and gender-influenced processes [22,23]. SD is strongly associated with certain health conditions and diseases, psychiatric disorders, medication or substance use, lack of knowledge, psychological or behavioral factors, relationship and cultural factors processes [23]. 

### 1.3. Study Background

More than 20 years after the Homeland War in Croatia (1991–1995), veterans still suffer from numerous health problems. Patients and/or health professionals may be reluctant to mention and discuss sexual symptoms [24], and a huge proportion of SD remains undiagnosed. Despite that, clinical observations and rising awareness have encouraged the recognition and assessment of SD in this patient group, and case reports [25] and research articles [26,27] regarding SD in veterans with PTSD in Croatia have been published. 

The main objective of this research is to systematically identify predictors of SD among veterans with PTSD. The main hypothesis of the study is that SD are predicted by overall PTSD symptom severity and by severity of D symptom cluster (Trauma-Related Negative Alterations in Cognition and Mood).

## 2. Experimental Section

### 2.1. Participants and Procedure

Participants were male war veterans (*N* = 300) recruited from a pool of patients referred to the Regional Center for Psychotrauma (RCP) and Department of Psychiatry within the Clinical Hospital Center (CHC) Rijeka, the Referral Center for PTSD of the Ministry for Health of the Republic of Croatia (*N* = 250), and the Daily Hospital for PTSD and Department of Psychiatry within the General Hospital (GH) Našice for treatment. Most of the veterans participated in operations on different and almost all battlefields. Thirteen of those whom we approached refused to participate, while two patients did not complete the questionnaires. 

Eligibility was determined by meeting diagnostic criteria for war-related PTSD as defined in DSM-5 [16]. Three patients were not eligible for the study as they did not meet the criteria for PTSD diagnosis. We continued recruiting patients until the number of 300 participants was reached. There were no differences in sociodemographic characteristics between those who refused to participate, those who did not complete the questionnaires, and those who were not eligible for the study.

The inclusion criteria for the study were: participation in the Homeland War as a soldier, experiencing at least one war-related traumatic event defined in the DSM-5 criteria for PTSD (personal experience of combat or exposure to a war zone), male gender, and age below 65. The exclusion criteria for the study were: active psychosis, moderate or high suicide risk measured by the Mini-International Neuropsychiatric Interview (MINI) for DSM-IV [28], and deformities, injury, or mutilation of the genital organs. None of the participants met the exclusion criteria.

Research consisted of two parts, i.e., a clinical interview and self-report questionnaires. The interviews were conducted by five psychiatrists and two psychologists from the two study sites. Sociodemographic data were collected through a questionnaire created for study purposes. The interviews and filling in of the questionnaires were usually completed in one or two sessions. The study was approved by the Ethics Committees of the Faculty of Medicine, University of Rijeka, CHC Rijeka, and GH Našice. Written informed consent was obtained from all participants after detailed information about the study was provided to them.

The study sample included a total of 300 male veterans. At the time of participation in the study, the majority of participants were in ambulatory treatment (66.8%), while other participants were involved in day-hospital treatment (19.3%) or club for PTSD (7.5%), or were hospitalized (6.4%). Table 1 provides further information on sample demographics.

### 2.2. Measures

#### 2.2.1. PTSD Checklist for DSM-5 (PCL-5) with Criterion A

The PCL-5 with Criterion A [29] is a self-report measure was revised to match the adapted DSM-5 criteria for PTSD. The interpretation of the PCL-5 should be made by a clinician. A PTSD diagnosis can be made provisionally considering items rated 2 = moderately or higher as a symptom endorsed according to the DSM-5 diagnostic rule (at least one B, one C, two D, and two E symptoms present). DSM-5 symptom cluster severity scores can be obtained by summing the scores for the items within a given cluster, i.e., cluster B (items 1–5), cluster C (items 6–7), cluster D (items 8–14), and cluster E (items 15–20). A total symptom severity score (range 0–80) can be obtained by summing the scores for each of the 20 items. Preliminary validation work was sufficient to make a cut-point score of 33, which was chosen for the purpose of this study [29]. Previous validation studies showed good psychometric properties for evaluating PTSD [30,31,32,33]. Cronbach’s alpha in our study for clusters of symptoms ranged from 0.67 to 0.85, and to 0.89 for total PCL-5. The Criterion A measure was included in the assesment according the criteria of DSM-5 [16].

#### 2.2.2. The International Index of Erectile Function (IIEF) 

IIEF [34] is a widely used, multi-dimensional self-report instrument for the evaluation of male sexual function over the last four weeks [34]. It consists of 15 questions grouped into five domains that assess erectile function (Q1,2,3,4,5,15), intercourse satisfaction (Q6,7,8), orgasmic function (Q9,10), sexual desire (Q11,12), and overall satisfaction (Q13,14). Each item is rated from 1 (very low; almost never or never; extremely difficult) to 5 (very high; almost always or always; not difficult). Scores for domains are calculated as the sum of the answers, with lower scores indicating worse functioning. The score for erectile function can be calculated and used to classify the severity of dysfunction as severe, moderate, mild, or no dysfunction. For other domains, a higher score indicates better function. The IIEF meets psychometric criteria for test reliability and validity, has a high degree of sensitivity and specificity, and correlates well with other measures of treatment outcome [34,35,36,37]. Cronbach’s alpha was 0.96 for erectile function, 0.91 for orgasmic function, 0.89 for sexual desire, and 0.91 for intercourse satisfaction and overall satisfaction. 

#### 2.2.3. Premature Ejaculation Diagnostic Tool (PEDT)

PEDT [38,39] is a self-report instrument for the evaluation of the presence and severity of premature ejaculation. Each PEDT item is rated from 0 (not difficult at all; almost never or never; not at all) to 4 (extremely difficult; almost always or always, extremely), with a higher score indicating more difficulties with premature ejaculation. Previous validation studies have shown satisfactory feasibility, reliability, and validity of the PEDT [38,39]. Cronbach’s alpha for PEDT scale in our study was 0.87.

#### 2.2.4. Male Sexual Dysfunction Criteria

The DSM-5 [16] classification recognizes four male sexual dysfunctions: delayed ejaculation (DE), erectile disorder (ED), male hypoactive sexual desire (HSD), and premature (early) ejaculation. To be diagnosed with SD, the symptoms must be present for at least six months, cause significant distress, and not be caused exclusively by a non-sexual mental disorder, significant relationship distress, medical illness, or medication. Also, these diagnoses are applicable to men who engage in non-vaginal sexual activity, but unfortunately, the specific duration criteria remain unknown [16]. For the purposes of this study, the following criteria were applied for a provisional diagnosis: DE—items Q9 or Q10 on IIEF rated 2 or less.ED—sum of scores on IIEF items Q1-Q5 and Q15 was 16 or less.HSD—items Q11 or Q12 on IIEF rated 2 or less.PE—item 2 on PEDT (*Do you ejaculate before you want to?*) rated 3 (Over half the time—>75%) or 4 (Always or Almost always—100%).

#### 2.2.5. Relationship Assessment Scale (RAS) 

The RAS [40,41] is a seven-item measure of global relationship satisfaction. Responses are on a five-point Likert scale, and either the total or the average score can be used in the interpretation. Average scores range from 1 to 5; total scores range from 7 to 35 (used in this study). Higher scores indicate greater relationship satisfaction. The reliability and validity of the English RAS have been established [41]. Cronbach’s alpha in our study was 0.87. 

#### 2.2.6. Mini-International Neuropsychiatric Interview (MINI)

Comorbid psychiatric disorders were diagnosed using the Croatian version of MINI for DSM-IV [28]. It is a brief and valid structured clinical interview meeting the need for a short but accurate structured psychiatric interview for multicenter clinical trials and epidemiology studies, to be used as a first step in outcome tracking in nonresearch clinical settings. This interview enables researchers to assess the 17 most common psychiatric disorders in DSM-IV. 

#### 2.2.7. Anatomical Therapeutic Chemical (ATC) Classification System

Self-reported data about drug consumption are classified in accordance with the ATC classification [42]. In brief, the ATC system classifies therapeutic drugs. The purpose of the system is to serve as a tool for drug utilization research in order to improve the quality of drug use. In the ATC classification system, the drugs are divided into different groups according to the organ or system on which they act and their chemical, pharmacological, and therapeutic properties. Drugs are classified into five different groups. 

### 2.3. Data Analysis 

#### 2.3.1. Data Analysis Plan 

The aim of the study was to assess the predictive models of several sexual dysfunctions in male veterans with PTSD. Average score of erectile function, orgasmic function, sexual desire, intercourse satisfaction, overall satisfaction (all measured by IIEF), and premature ejaculation (measured by PEDT) were the outcome variables. Prediction variables were characteristics identified as relevant for sexual dysfunction in previous studies. Two sets of hierarchical regression analyses were executed for each of the sexual functions (one without and one with relationship satisfaction) in order to assess the best models for the overall sample of veterans with PTSD and for the subset of veterans in relationship. In order to control for covariances, predictor variables were entered in the following steps/models: (1) sociodemographic variables, (2) comorbid disorders (psychiatric and others), (3) medication used (psychotropic and other drugs), (4) variables related to PTSD (deployment duration and PTSD symptoms), and (5) relationship satisfaction (subset sample of veterans in relationship). The exclusion criterion for dichotomous predictors was set to 10 or less events per variable [43]. The inclusion criterion for a prediction variable was a significant association with the outcome variable. 

#### 2.3.2. Statistical Analysis

Statistical analysis was performed with Statistica software, version 12 (Dell Inc. Inc., Tulsa, OK, USA). Data are presented as *N* (%) or M (sd). Chi-square tests for categorical variables and independent sample *t*-tests for continuous variables were used to compare veterans with or without provisional diagnosis of sexual dysfunction. Pearson and Spearman correlation coefficients were calculated between sexual functions and the variables of interest. Several hierarchical multiple regressions were performed to test for the best prediction models for the outcome variables of erectile function, orgasmic function, sexual desire, intercourse satisfaction, overall satisfaction, and premature ejaculation. All models were controlled for basic assumptions. Two issues with multicollinearity were encountered, i.e., between cluster D and cluster E symptoms with overall PTSD symptoms, and between in-a-relationship status and relationship satisfaction in the subset sample. Overall PTSD symptoms were excluded from both sets of samples, and the in-a-relationship status variable from the subset sample. Missing values were controlled for listwise. Probability significance was set to *p* ≤ 0.05.

## 3. Results

### 3.1. Sociodemographic Data

Sociodemographic data for the overall sample are presented in Table 1. 

### 3.2. Trauma Exposure and PTSD

The average duration of active participation in the Homeland war was 30 (19.516) months, ranging from 1 month to 70 months. 

Twenty-three percent of participants had sought psychiatric help in the period from 1991 to 1995, while the war was ongoing. The average intensity for overall PTSD symptoms was 57.5 (10.92) within the range of 33 to 80. The average intensity of B symptoms was 15 (3.25), of C symptoms 6.2 (1.47), of D symptoms 18.7 (5.15), and of E symptoms 17.7 (3.90).

### 3.3. Prevalence of SD and Association with Sociodemographic Data and PTSD

The average score for erectile function was 16 (9.71), which relates to moderate dysfunction. The average score for orgasmic function was 5.8 (3.31) (theoretical maximum = 8), for sexual desire 5.8 (2.47) (theoretical maximum = 8), for intercourse satisfaction 6.51 (4.71) (theoretical maximum = 12), and for overall satisfaction 6.3 (2.44) (theoretical maximum = 8). The average score for PEDT was 7.43 (5.14) within the range of 0 to 20. 

According to provisional criteria for male sexual dysfunction (described in methodology), the following rates were found: DE 124 (44%, *n* = 282), ED 134 (46.2%, *n* = 290), HSD 128 (44.6%, *n* = 287), and PE 59 (21.3%, *n* = 277). Overall, on the basis of self-reported data, 98 (35.1%) of veterans with PTSD did not meet, while 181 (64.9%) participants met provisional criteria for at least one male SD in the last month. Out of possible four SD, one SD had 49 (17.6%) participants, two SDs had 36 participants (12.9%) participants, three SD had 80 participants (28.7%), and four SD had 16 participants (5.7%). 

As presented in Table 1, participants in a relationship and participants with medium income were less likely to have a provisional diagnosis of SD. Participants who met the provisional diagnosis of SD were significantly less satisfied with their relationship compared to participants without SD. Veterans with SD had significantly greater severity of cluster D, cluster E, and overall symptoms of PTSD. They did not differ for duration of deployment or for cluster B and cluster C symptoms.

Prevalence of comorbid disorders and drug use and association with SD are presented in Appendix A. 

### 3.4. Prediction Models of Sexual Dysfunctions among War Veterans with PTSD

Predictor variables for each model (i.e., sexual function) were selected on the basis of the following criteria: variables with events greater than 10 and significant correlation with the outcome variable (Appendix A). However, some variables were included regardless, such as age and all clusters of PTSD symptoms. Also, analysis showed great correlation coefficients between overall PTSD symptoms intensity and cluster D and E symptom intensity (variance inflation factor (VIF) > 8). Because of the multicollinearity issues, overall PTSD symptoms were not included in the models. The variable “in a relationship” had high multicollinearity with relationship satisfaction (VIF > 8), and, therefore, only relationship satisfaction was included in the models for the subset of veterans in a relationship. The final steps for all tested models are presented in Appendix A. An overview of individual significant predictors for each sexual function is given in Table 2 and Table 3.

#### 3.4.1. Erectile Function

The initial model tested for erectile function included age, low income, medium income, not married, married, “in a relationship” status (Model 1: *R*^2^ = 0.134, *F* = 7.060, *p* < 0.001). In the second step ongoing major depressive episode (MDE), panic disorder lifetime, essential hypertension, and hyperplasia of prostate (Model 2: *R*^2^ = 0.181, *F* = 5.816, *p* < 0.001) were included; in the third step, use of antidepressants, hypnotics, and sedatives (Model 3: *R*^2^ = 0.184, *F* = 5.514, *p* < 0.001) was added; in the fourth step, war deployment in months and cluster B, C, D, and E symptoms were added (model 4: *R*^2^ = 0.257, *F* = 5.651, *p* < 0.001). Significant predictors did not change through the models. The final model explained 25.7% of the variance of erectile function. Variables with significant independent contribution were being in a relationship, having essential hypertension, and severity of D cluster symptoms (Table 2). 

In the subset of participants in a relationship, Model 1, containing age, low income, medium income, not married, and married, was not significant, since the variable relationship status was removed. Model 2 accounted for 9.3% (*F* = 2.200, *p* = 0.019), Model 3 for 9.98% (*F* = 2.097, *p* = 0.22), and Model 4 for 20.1% (*F* = 3.273, *p* < 0.001) of the variance of erectile function. The final model with relationship satisfaction added explained 27.9% of the variance of erectile function (*F* = 5.457, *p* < 0.001). Significant individual predictors were having essential hypertension, severity of cluster D symptoms, and relationship satisfaction. (Table 3).

#### 3.4.2. Orgasmic Function

The initial model tested for orgasmic function included age, higher education, low income, medium income, married, and “in a relationship” status (Model 1: *R*^2^ = 0.100, *F* = 4.619, *p* < 0.001). In the second step, ongoing MDE, panic disorder lifetime, essential hypertension, hyperplasia of prostate, and disorders of lipoprotein metabolism were added (Model 2: *R*^2^ = 0.165, *F* = 4.397, *p* < 0.001); in the third step, use of antidepressants, hypnotics, and sedatives (Model 3: *R*^2^ = 0.189, *F* = 4.751, *p* < 0.001) was included; in the fourth step, cluster B, C, D, and E symptoms (Model 4: *R*^2^ = 0.248, *F* = 4.628, *p* < 0.001) were added. Higher education level was a significant individual contributor until psychotropic medication was introduced in the third step. The final model explained 24.8% of the variance of orgasmic function. Significant independent predictors were being in a relationship, use of antidepressants, having hypertension, and severity of cluster D symptoms (Table 2). 

In the subset of participants who were in a relationship, Model 1 was not significant and accounted for 5.4% variance of orgasmic function. Model 2 (*R*^2^ = 0.121, *F* = 2.601, *p* = 0.004), Model 3 (*R*^2^ = 0.152, *F* = 3.098, *p* < 0.001), and Model 4 (*R*^2^ = 0.244, *F* = 3.839, *p* < 0.001) were all significant. The final model explained 29.5% of the variance of orgasmic function (*F* = 4.679, *p* < 0.001). Significant individual predictors were: use of antidepressants, presence of essential hypertension, severity of cluster D symptoms, and relationship satisfaction (Table 3). There was no significant change in the significance of individual predictors through the models. 

#### 3.4.3. Sexual Desire

The initial model for sexual desire included age, low and medium income, and “in a relationship” status (Model 1: *R*^2^ = 0.055, *F* = 4.131, *p* = 0.003). In the second model, alcohol use disorder (AUD) was added (Model 2: *R*^2^ = 0.074, *F* = 4.521, *p* = 0.001); in the third model, antidepressant use was included (Model 3: *R*^2^ = 0.095, *F* = 4.901, *p* < 0.001); in the fourth model, cluster B, C, D, and E symptoms (Model 4: *R*^2^ = 0.166, *F* = 5.482, *p* < 0.001) were added. All the models were significant, and there was no change in the significance of individual predictors. The final model explained 16.6% of the variance of sexual desire in the entire sample. Predictors with independent contribution were being in a relationship, presence of an AUD, use of antidepressant, and severity of cluster D symptoms (Table 2).

In the subset sample of veterans in a relationship, the sociodemographic variables entered did not significantly contribute to the variance of sexual desire (Model 1: *R*^2^ = 0.035, *F* = 2.219, *p* = 0.068) Addition of AUD in step two (Model 2: *R*^2^ = 0.054, *F* = 2.789, *p* = 0.018), antidepressant in step three (Model 3: *R*^2^ = 0.081, *F* = 3.536, *p* = 0.002), and clusters of PTSD symptoms in step four (Model 4: *R*^2^ = 0.172, *F* = 4.930, *p* < 0.001) significantly increased the variance of sexual desire. The final model which included relationship satisfaction explained 19.6% of sexual desire in veterans in a relationship (*F* = 5.227, *p* < 0.001). As in the total sample, predictors with significant independent contribution were use of an antidepressant and severity of cluster D symptoms, but not AUD. A significant contributor was also relationship satisfaction (Table 3). The significant predictors did not change through the models.

#### 3.4.4. Intercourse Satisfaction (IS)

The initial model for the intercourse satisfaction consisted of age, low income, medium income, not married, divorced, married and relationship status (Model 1: *R*^2^ = 0.139, *F* = 6.748, *p* < 0.001). In the second model, ongoing MDE, other anxiety disorders, essential hypertension, and hyperplasia of prostate were added (Model 2: *R*^2^ = 0.168, *F* = 5.287, *p* < 0.001); in the third (final) model, war deployment in months, cluster B, C, D, and E symptoms (Model 3: *R*^2^ = 0.251, *F* = 5.714, *p* < 0.001) were included. There was no change in individual predictors through the models, and the final model explained 25.1% of intercourse satisfaction in veterans with PTSD. The identified significant predictors were being in a relationship, presence of essential hypertension, and severity of cluster D symptoms (Table 2). 

In the subset sample of veterans who were in a relationship, the final model accounted for 31.6% of intercourse satisfaction (*F* = 7.382, *p* < 0.001). All tested models were significant (Model 1: *R*^2^ = 0.067, *F* = 2.982, *p* = 0.008; Model 2: *R*^2^ = 0.102, *F* = 2.788, *p* = 0.003; Model 3: *R*^2^ = 0.188, *F* = 3.987, *p* < 0.001). The significance of predictors did not change through the models. In contrast to the overall sample, essential hypertension was not a significant predictor of IS among veterans in a relationship. Severity of D cluster symptoms and relationship satisfaction were independent significant contributors (Table 3). 

#### 3.4.5. Overall Satisfaction 

In the first model for overall satisfaction, the following variables were entered: age, low income, medium income, and “in a relationship” status (Model 1: *R*^2^ = 0.061, *F* = 4.474, *p* = 0.002). In the next step, comorbid diseases, ongoing MDE, panic disorder lifetime, other anxiety disorders, AUD, essential hypertension, and hyperplasia of prostate were entered (Model 2: *R*^2^ = 0.140, *F* = 4.371 *p* < 0.001); in the third step, use of antidepressants (Model 3: *R*^2^ = 0.145, *F* = 4.125, *p* < 0.001) was included; in the fourth step, clusters B, C, D, and E symptoms (Model 4: *R*^2^ = 0.210, *F* = 4.652, *p* < 0.001) were added. Recurrent panic disorder was a significant predictor until PTSD symptoms were entered in the last step. The final model explained 21% of the variance of overall satisfaction in veterans with PTSD. Significant individual predictors of overall satisfaction were being in a relationship, presence of an AUD, presence of essential hypertension, and severity of cluster D symptoms (Table 2). 

All the models tested for overall satisfaction among veterans with PTSD who were in a relationship were significant (Model 1: *R*^2^ = 0.038, *F* = 3.188, *p* = 0.024; Model 2: *R*^2^ = 0.117, *F* = 3.457, *p* < 0.001; Model 3: *R*^2^ = 0.136, *F* = 3.717, *p* < 0.001; Model 4: *R*^2^ = 0.215, *F* = 4.570, *p* < 0.001). The “other anxiety disorders” variable was a significant predictor until PTSD symptoms were entered in the fourth step. The final model in the subset sample explained 38.4% of the variance of overall satisfaction (*F* = 9.651, *p* < 0.001). Significant individual predictors were: presence of hyperplasia of prostate, use of an antidepressants, severity of cluster D symptoms, and relationship satisfaction (Table 3). It is important to note that relationship satisfaction by itself (β = 0.435) explained most of the variance of overall sexual satisfaction. 

#### 3.4.6. Premature Ejaculation

In the first model of premature ejaculation in the overall sample, the following sociodemographic variables were entered: age, not married, married, and “in a relationship” status (Model 1: *R*^2^ = 0.054, *F* = 3.779, *p* < 0.01). In Model 2, diabetes mellitus (DM) was added (*R*^2^ = 0.067, *F* = 3.795, *p* < 0.01), and cluster B, C, D, and E symptoms were added in Model 3. Significant individual predictors were DM and severity of cluster D symptoms (Table 2). 

Similar findings were reported in the subset sample of veterans in a relationship, as the final model contributed to 10.7% of the variance of premature ejaculation (*F* = 3.019, *p* < 0.001) with the independent significant contributors DM and cluster D symptoms (Table 3). Model 1, containing sociodemographic variables (*R*^2^ = 0.016, *F* = 1.257, *p* = 0.290), and Model 2 (*R*^2^ = 0.035, *F* = 2.079, *p* = 0.054), containing comorbid diseases, did not contribute significantly to the variance of premature ejaculation. Model 3, which included clusters of PTDS symptoms, was significant (*R*^2^ = 0.104, *F* = 3.277, *p* < 0.001). Relationship satisfaction added in the final model did not alter significantly the variance explained. 

## 4. Discussion

To the best of our knowledge, the present study is the first to suggest patterns of association of PTSD with different types of SD and to determine the predictors of this relationship. The results of the study support the main hypothesis that SD in veterans with PTSD are predicted by the severity of the D cluster of PTSD symptoms. The second part of the hypothesis that states SD are predicted by overall PTSD symptom severity is partially supported. We found that veterans with SD had significantly higher PTSD symptom scores than veterans without SD. Furthermore, overall PTSD symptom severity was significantly correlated with all types of SD (DE, ED, HSD, and PE) as well as intercourse satisfaction (IS) and overall satisfaction (OS). Analysis revealed significant multicollinearity of this predictor with D symptoms of PTSD, which implies that the association of PTSD symptom severity with SD is mediated and mostly depends on the quantity and severity of trauma-related negative alterations in cognition and mood. Previous studies found high rates of SD among male veterans with PTSD [1,2,3,4,5]. The results of our study are consistent with the scarce but increasing body of research that indicates that the severity of PTSD measured by overall scores on PTSD scales is not a significant predictor of SD in veterans with PTSD [2,5,11,12]. 

Beside the prevalence and correlation of SD with PTSD, it is important to understand the background of this relationship. A high score of D symptoms (Trauma-Related Negative Alterations in Cognition and Mood) appears to be the most prevalent predictor of SD among veterans with PTSD, emerging as a significant predictor of all types of SD (DE, ED, HSD, PE) as well as of IS and OS. D cluster includes three new symptoms according to the DSM-5 classification: negative expectations of self, others, or the world (replacing the sense of foreshortened future), persistent distorted blame of self or other for trauma, and pervasive negative emotional state. The presence of these symptoms and/or other symptoms from the D cluster, such as diminished interest or participation in significant activities, a feeling of detachment or estrangement from others, or a persistent inability to experience positive emotions, precludes a person’s capacity to engage adequately in sexual behavior(s). As a result, D symptoms predict lower levels of satisfaction in sexual life. The current DSM-5 classification embraces the four-factor model, as it provides a better representation of PTSD’s latent structure than the tripartite model of DSM-IV [43,44,45,46], which has received extensive criticism [47]. Our findings in veterans are consistent with prior research demonstrating that avoidance/numbing symptoms of PTSD are strongly linked to self-reported problems in sexual functioning. Nunnink and colleagues found that self-reported symptoms of emotional numbing predicted a greater likelihood of endorsing sexual problems [11]. The results of another study that investigated predictors of ED and self-reported sexual problems among 150 male combat veterans seeking outpatient treatment for PTSD revealed, beside various demographic, physical, and psychosocial risk factors, a significant zero-order correlation between avoidance/numbing symptoms and SD [2].

Partner relationship is the next prominent predictor of SD in veterans with PTSD. Results in the overall sample revealed that being in a partner relationship reduces the risk of DE, ED, HSD, IS, and OS. Being in a relationship has no predictive value for PE. Analysis in the sample of participants who were in a partner relationship indicated that a low level of relationship satisfaction was a significant predictor of DE, ED, HSD, and IS and OS. Relationship satisfaction was not a significant predictor of PE. The association of PTSD with impairments in romantic relationship satisfaction has been previously reported [11,19,20]. A recent meta-analysis of 23 studies found an association between the emotional numbing and avoidance symptom cluster and parent, child, family, and marital/partner functioning problems [48]. Sexual functioning and relationship satisfaction are also robustly, positively correlated in many different samples across a variety of adult populations, including those who are dating [49,50], in long-term relationships [51], and married [52,53]. A lower level of relationship satisfaction in our study sample was an independent predictor of SD and was not mediated by the severity of any PTSD cluster. Sexual functioning is one of the essential domains of relationship functioning. Association between SD and quality of relationship is bidirectional and reciprocal. Relationship problems caused by family stressors, economic reasons, lifestyle, etc. inevitably affect sexual functioning. Problems in sexual functioning may have an impact on all other domains of a relationship. In the context of PTSD, the quality of a relationship also depends on the accommodation capacities of the partner for mutual acceptance, which is important for healthy sexual functioning. Additionally, PTSD may affect relationship and sexual functioning indirectly through changes of behavioral patterns. For example, insomnia and nightmares are less likely to have a direct impact on sexual functioning than numbing symptoms. On the other hand, these symptoms may lead to sleeping in separate beds, allowing or encouraging the rituals and avoidant behavior that lessen the quality of a relationship and sexual functioning. This finding implies that therapeutic efforts directed to promoting relationship satisfaction in veterans with PTSD could have a positive effect on sexual functioning in most of its domains. Interestingly, being in a relationship and relationship satisfaction are not significant predictors of PE. This finding could be explained by considering PE symptoms as more of an individual than a relational problem, which in turn is not worsened or maintained by disturbances in a partner relationship.

Antidepressant use is a significant predictor of the impairment of orgasmic functioning and sexual desire, i.e., veterans with PTSD that use antidepressants have increased risk for DE and HSD. Surprisingly, antidepressant utilization did not show predictive values for ED and OS. Adverse sexual effects are frequent with commonly prescribed psychotropic drugs and are usually underestimated [24,54]. The recent clinical guidelines highlight antidepressants as first-line pharmacotherapeutic agents in the management of PTSD [55,56]. In spite of increasing rates of drug utilization (80%) among veterans with PTSD [57], some studies revealed a marked inconsistency with the current guidelines for treatment of PTSD, particularly in the post-conflict settings [58]. In that context, our finding of antidepressant use as a significant predictor of DE and HSD is important, bearing in mind that 41.5% of our participants have DE and 45.4% have HSD. The findings are consistent with a meta-analysis which revealed increased rates of SD among patients in treatment with antidepressants [54]. Furthermore, higher rates of total and specific-treatment emergent SD and specific phases of dysfunction were found for drugs with a predominantly serotonergic action, including selective serotonin reuptake inhibitors (SSRIs) and serotonin and norepinephrine reuptake inhibitors (SNRIs) [55,59]. Ejaculation-delaying effect of antidepressants on orgasmic function is, on the other hand, the basis for the use of either tricyclic antidepressant or SSRIs in treatment of PE. Among other medications from this pharmacological group, paroxetine has the most prominent ejaculation-delaying effect [60] caused by its impact on serotonergic receptors, cholinergic receptor blockade, and inhibition of nitric oxide synthase [61,62,63,64]. It is also supported by the results of this study, as antidepressants are not significant predictors of PE. 

Arterial hypertension was a significant predictor of ED, DE, IS, and OS in the overall sample. It was a significant predictor of ED and DE in the sample of veterans in a relationship. These findings are consistent with those of numerous studies that emphasize high blood pressure as a risk factor for SD [65,66,67]. Actually, vasculogenic ED is considered part of a systemic vasculopathy and has a known relationship with cardiovascular risk factors such as hypertension, diabetes, dyslipidemia, and smoking [68]. A research that included 1255 male participants revealed that lower systolic and diastolic blood pressure were associated with better sexual functioning [67]. 

The significant predictor of PE in the overall sample and among participants in a relationship was DM. Patients with DM have higher rates of various SD directly related to the deleterious complications of their disease. [69,70,71]. DM is also indirectly related to SD through anxiety and depression that are often experienced by men with DM [72]. Of these, ED was most commonly reported [69,70,71]. Some studies reported higher rates of PE in patients with DM, indicating duration, severity, and poor metabolic control as the main risk factors for PE in diabetic patients. On the other hand, a close relationship between ED and PE exists. Some authors suggest that the longer the erectile problem, the worse the anxiety, and the more marked the PE [73]. Because of performance anxiety regarding their erectile reliability, patients could rush through an intercourse, with PE as a deleterious consequence [74].

AUD was a significant predictor of SD and OS in the overall sample and in those who patients were in a relationship. The results are consistent with the findings of a previous research and meta-analysis [65]. This finding is important in the context of populations of veterans with PTSD, as repeated heavy drinking is one of the common strategies to alleviate trauma symptoms that may lead to the development of AUD. The prevalence of AUD in PTSD is also high. For example, in the US, 42% of PTSD subjects met criteria for AUD diagnosis [75]. The prevalence of alcohol-induced sexual dysfunction is unclear, probably because of underreporting. Sexual disorders ranging from 8% to 95.2% have been reported in men with chronic alcohol use [76,77,78,79,80]. The common dysfunctions reported were lack of sexual desire [79,80], premature ejaculation [81,82], and erectile dysfunction [76,82,83,84].

Although the prevalence of the SD was not the main focus of this research, it is indicative that none of the participants reported being diagnosed with SD. Only one veteran with PTSD reported the utilization of a medication prescribed to treat ED (sildenafil). This finding is completely inconsistent with data from previous studies suggesting that SD is strongly related to PTSD, particularly war-related PTSD [1,2,3,4,5]. A backup check of medical records confirmed only one diagnosis of SD recorded in the study sample. Two widely used instruments for the assessment of the presence and severity of the different types of SD were applied with restrictive criteria for severity of SD symptoms, consistent with DSM-5 for diagnosis of SD (i.e., present in at least 75% of sexual activity occasions) in order to avoid over-diagnosing minor and potentially transient problems in sexual functioning. According to that criteria, the following rates were found: SD in 64.9% of patients, DE in 44%, ED in 46.2%, HSD in 44.6%, PE in 21.3%. The rates of SD differ across studies [6,7,8,9,10], mainly because of different methodological approaches. Predominantly, two methods for identifying SD have been used in research. In some studies, the estimation of SD diagnosis was based on reported patients’ symptoms and problems in sexual functioning, with wide criteria for SD applied. In another study, the presence of SD was considered if SD diagnosis was recorded or medication for SD was used, which may be a more conservative approach. Both methods for identifying SD may be problematic. If we chose the second approach, we could conclude that veterans with PTSD in our sample had superior sexual functioning. Therefore, we chose the first approach, bearing in mind that self-reported symptoms in questionnaires can be used only for an estimation and provisional diagnosis of SD. Clinical interviews are irreplaceable and necessary to sufficiently diagnose SD if they are conducted by well-trained personnel, who are also trained about social stigmatization. Conversely, they may contribute to underreporting biases arising from personal concerns about social stigmatization and lack of privacy, particularly in older or less educated participants [85]. The rates of SD in this study confirm that the complete absence of SD diagnosis in our clinical setting could not be a consequence of non-clinically significant problems among veterans. A dramatically higher self-reported prevalence of SD suggests a number of veterans may be choosing not to disclose problems in sexual functioning with their healthcare providers because of embarrassment, discomfort, or lack of knowledge about treatment possibilities. 

### 4.1. Strenghts

This study was primarily designed to assess SD in the population of veterans with PTSD. Veterans with PTSD were included regardless of their relationship status, as even those not in a current romantic relationship may engage in sexual behavior and are often overlooked in studies. Data related to military deployment, sociodemographic and relationship factors, psychiatric comorbidity, psychotropic and other medication, and medical conditions were systematically collected, as all these factors could be important contributors to SD. PTSD symptoms were assessed jointly, but, more importantly, the impact of each cluster of PTSD symptoms (according to DSM-5 classification) on sexual functioning was also assessed. In assessing SD, we applied a comprehensive approach covering a broader range of possible sexual health problems as well as perceived sexual satisfaction. 

### 4.2. Limitations

This study has several limitations. Findings from this study may not be reflective of and generalisable to the broader veteran or nonveteran population. Because of the many variables tested, data analysis suffered from multiple comparisons, allowing for possible false positive effects/predictors. Health-care-seeking participants could suffer from more serious problems in each area covered by the research. Furthermore, the generalisability is limited by a gender-imbalanced sample, as only male veterans were included in the research. Because of the cross-sectional design, the temporality of the relationship between the different studied variables and sexual dysfunction could not be evaluated. The findings were based on self-reported symptoms from questionnaire measures. Self-reports of sexual activity and satisfaction may be under- or overreported because of stigmatization.

## 5. Conclusions

One of the most salient implications of the current study is the importance of sexual health assessment in veterans with PTSD. This study represents an advancement in our currently limited understanding of patterns of association of PTSD with different types of SD and of the predictors of that relationship. As veterans with PTSD are more likely to suffer from SD if they experience more D symptoms and if they are not in a relationship or are less satisfied with the relationship, future research should develop therapeutic interventions more focused on the negative appraisals, emotional numbness, and irritability and other negative cognitions and emotions, as well as interventions intended to improve relationship functioning with the aim to lessen the rates of SD in this population. Psychotherapy is strongly recommended as the first-line treatment approach in PTSD. Sex therapy is effective in the variety of the SD, and couple psychotherapy is an established approach for relationship problems and dissatisfaction. Psychotherapeutic treatments, which would comprehensively cover different aspects of the problems in patients with PTSD and SD comorbidity, could have greater compliance rates, less iatrogenic adverse effects, and better treatment effects. Psychotropic treatment with fewer adverse sexual effects and management of the treatment-emergent side effects are of utmost importance if pharmacotherapy is applied. Medical conditions, particularly those stress-related and frequent in study populations with diabetes and hypertension, carry an additional burden of increased risk for SD. Appropriate prevention, screening for those conditions, and their active treatment could improve the sexual life of veterans with PTSD. 

## Figures and Tables

**Table 1 jcm-08-00432-t001:** Sociodemographic characteristic and differences according to the presence of sexual dysfunction.

	All *n* = 300	Sexual Dysfunction NO *n* = 98	Sexual Dysfunction YES *n* = 181	Statistics	Probability
	M(sd) or *N*(%)	M(sd) or *N*(%)	M(sd) or *N*(%)		
Age (years)	52.4 (5.82)	52.4 (5.15)	52.1 (5.74)	*t* = 0.470	0.639
**Marital Status (yes)**					
Married	197 (65.7%)	68 (57.4%)	116 (63%)	*χ*^2^ = 0.795	0.373
Cohabitation	18 (6%)	5 (31.2%)	11 (68.8%)	*χ*^2^ = 0.112	0.738
Divorced	33 (11%)	9 (29%)	22 (71%)	*χ*^2^ = 0.568	0.451
Widower	2 (0.7%)	--	--	--	
Not married	37 (12.3%)	9 (27.3%)	24 (72.7%)	*χ*^2^ = 1.013	0.314
Other	12 (4%)	7 (58.3%)	5 (41.7%)	*χ*^2^ = 2.964	0.085
In a relationship (yes)	259 (86.3%)	94 (38.8%)	148 (61.2%)	*χ*^2^ = 11.067	0.001
**Financial Status (yes)**					
Low income	96 (32.7%)	25 (27.5%)	66 (72.5%)	*χ*^2^ = 3.471	0.062
Medium income	186 (63.3%)	71 (39.7%)	75 (60.3%)	*χ*^2^ = 4.516	0.034
High income	12 (4.1%)	2 (22.2%)	7 (77.8%)	*χ*^2^ = 0.679	0.410
**Education (yes)**					
Elementary	39 (13.1%)	14 (40%)	21 (60%)	*χ*^2^ = 0.417	0.518
Secondary	231 (77.5%)	77 (34.8%)	144 (65.2%)	*χ*^2^ = 0.038	0.846
Higher	26 (8.7%)	7 (30.4%)	16 (69.6%)	*χ*^2^ = 0.242	0.697
**In the Last Month ^1^ (yes)**					
Alcohol	33 (89%)	6 (20.7%)	23 (79.3%)	*χ*^2^ = 2.960	0.085
Cigarettes	50 (16.7%)	21 (42.9%)	28 (57.1%)	*χ*^2^ = 2.837	0.092
Marijuana	4 (1.3%)	--	--		
War deployment (months)	29.6 (19.2)	29.6 (19.34)	30 (19.51)	*t* = −0.163	0.871
Cluster B symptoms	15 (3.25)	15.1 (2.92)	14.9 (3.32)	*t* = 0.534	0.594
Cluster C symptoms	6.2 (1.47)	6.1 (1.48)	6.3 (1.46)	*t* = −1.047	0.296
Cluster D symptoms	18.7 (5.15)	17.2 (4.81)	19.4 (5,06)	*t* = −3.612	<0.001
Cluster E symptoms	17.7 (3.9)	16.9 (4.27)	18 (3.64)	*t* = −2.422	0.016
Total PTSD symptoms	57.5 (10.92)	55.2 (10.46)	58.6 (10.71)	*t* = −2.559	0.011
Relationship satisfaction	25.9 (6.37) ^1^	28.1 (5.15) ^2^	24.8 (6.01) ^3^	*t* = 4.298	<0.001

^1^*n* = 256, range 7–35; ^2^
*n* = 148; ^3^
*n* = 98.

**Table 2 jcm-08-00432-t002:** Overview of individual significant predictors in the final step of hierarchical regression analysis for the overall sample.

	Erectile Function		Orgasmic Function		Sexual Desire		Inter-Course Satisfaction		Overall Satisfaction		Premature Ejaculation	
	B	β	B	β	B	β	B	β	B	β	B	β
In a relationship	7.69	**0.28 ****	1.98	**0.21 ***	0.93	**0.13 ***	3.94	**0.30 ****	1.17	**0.15 ***	1.99	0.13
Alcohol use dis. ^1^					−1.62	**−0.13 ***			−1.47	**−0.12 ***		
Diabetes mellitus											1.71	**0.12 ***
Hypertension, esse. ^2^	−3.99	**−0.20 ****	−0.98	**−0.15 ***			−1.49	**−0.15 ****	−0.71	**−0.14 ***		
Antidepressant	−1.3	−0.06	−1.12	**−0.15 ****	−0.71	**−0.14 ***			−0.36	−0.07		
Cluster D symptoms	−0.46	**−0.24 ****	−0.15	**−0.23 ****	−0.14	**−0.29 ****	−0.23	**−0.25 ****	−0.11	**−0.23 ****	0.21	**0.21 ***

* *p* ≤ 0.05; ** *p* ≤ 0.01; ^1^ Alcohol use disorders; ^2^ Hypertension, essential; significant values are in bold.

**Table 3 jcm-08-00432-t003:** Overview of individual significant predictors in the final step of hierarchical regression analysis for the subset sample of veterans in a relationship.

	Erectile Function		Orgasmic Function		Sexual Desire		Inter-Course Satisfaction		Overall Satisfaction		Premature Ejaculation	
	B	β	B	β	B	β	B	β	B	β	B	β
Diabetes mellitus											1.91	**0.14 ***
Hypertension, esse. ^1^	−3.08	**−0.17 ****	−0.78	**−0.12 ***			−0.84	−0.09	−0.48	−0.10		
Hyperplasia prost. ^2^	−2.88	−0.05	−169	−0.08			−2.19	−0.08	−1.99	**−0.13 ***		
Antidepressant	−1.17	−0.06	−0.96	**−0.15 ***	−0.73	**−0.15 ***			−0.47	**−0.11 ***		
Cluster D symptoms	−0.38	**−0.21 ****	−0.15	**−0.24 ****	−0.14	**−0.30 ****	−0.11	**−0.23 ****	−0.09	**−0.19 ****	0.23	**0.23 ****
Relationship satisf. ^3^	0.43	**0.29 ****	0.14	**0.28 ****	0.06	**0.16 ****	0.27	**0.39 ****	0.16	**0.44 ****	0.05	0.07

* *p* ≤ 0.05; ** *p* ≤ 0.01; ^1^ Hypertension, essential; ^2^ Hyperplasia of prostate; ^3^ Relationship satisfaction; significant values are in bold.

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
