# Peer review of "Predictors of Sexual Dysfunction in Veterans with Post-Traumatic Stress Disorder"

_jcm, 2019, doi:10.3390/jcm8040432_

Reviewer 1 Report

The authors report from their cross-sectional study investigating 300 male veterans with PTSD in the mean age of 52.4 years, to identify predictors of sexual dysfunction in PTSD by several hierachical multiple regression analyses. They observed severity of ‘PTSD cluster D symptoms’ and ‘relationship status/satisfaction’ as significant individual predictors for sexual dysfunction in this sample.

The topic of this study is interesting and of great clinical relevance. However, I have some concerns that should be addressed.

Considering that the readers of the JCM may not be familiar with psychiatric / psychometric measures, the authors should consider to provide full names rather than abbreviations for the conducted assessments in the abstract. If this conflicts with the limited word count, they should consider to mention the domains they assessed (e.g. erectile function, premature ejaculation,…). Similar, the authors should provide an explanation of the so-called ‚cluster D symptoms‘ for the readers of JCM in the abstract and they should specified with regard to their results.

The introduction should be strengthened to the scope of their investigation. By elaborating the predictors of sexual dysfunction in general population, the authors may motivate their assessment of somatic diagnoses and/or medication (including psychotropic). Nevertheless, this part could be shortened since it heavily distracts from the scope of the study. 

The authors declare that sexual dysfunction is predicted by overall PTSD symptom severity and by the severity of D symptoms as main hypothesis. However, this main hypothesis (in particular regarding cluster D symptoms) is not motivated in the introduction section and should be elaborated.

Similar, the participants and procedure paragraph should be shortened and strengthened, e.g. the information that CHC Rijeka is providing medical care for 600000 inhabitants does not add any relevant information to the scope of the study. The information in lines 154 to 163 are also presented in table 1. Further, the authors should motivate the inclusion of the LEC-5 ? What does this measure contribute to the scope of the study since a diagnosis of PTSD implicates a traumatic event and this measure is not mention further?

What does the authors mean by ‘preliminary validation work is sufficient to make cut-point score of 33 which is chosen for the purpose of this study’ and then, restrict their statement by mentioning the variety of cut-off scores ? This is confusing and the authors should instead provide a reference for the cut-point score.

In my opinion, there is no need to introduce the ICD-10 classification. This paragraph also does not add anything to the manuscript since somatic diagnoses were solely assessed by self-reports.

I have concerns regarding the statistical inference considering the several statistical tests / correlations conducted. Could the authors please make a statement on multiple comparison procedures ?

The presentation of the results is someway confusing due to the amount of variables in the tables. The authors should consider to present solely relevant / significant results and provide the other results in the supplementary material section.

In sum, apart from statistical inference, I am struggling with the manuscript since many informations are included that distract from the main results.

Author Response

Dear Reviewer 1,

We appreciate the opportunity to resubmit our article. We described how we have responded, point by point, to your comments. We also indicated where changes were made in the manuscript (Comments) to accommodate your requests.  We provide the article with track changes and another copy with accepted changes. Accompanying this letter, please find a detailed responses.

Best regards,

Point by Point Reply to Review Report (Reviewer 1)

The authors report from their cross-sectional study investigating 300 male veterans with PTSD in the mean age of 52.4 years, to identify predictors of sexual dysfunction in PTSD by several hierarchical multiple regression analyses. They observed severity of ‘PTSD cluster D symptoms’ and ‘relationship status/satisfaction’ as significant individual predictors for sexual dysfunction in this sample.

The topic of this study is interesting and of great clinical relevance. However, I have some concerns that should be addressed.

1.     Considering that the readers of the JCM may not be familiar with psychiatric / psychometric measures, the authors should consider to provide full names rather than abbreviations for the conducted assessments in the abstract. If this conflicts with the limited word count, they should consider to mention the domains they assessed (e.g. erectile function, premature ejaculation,…). Similar, the authors should provide an explanation of the so-called ‚cluster D symptoms‘ for the readers of JCM in the abstract and they should specified with regard to their results. 

Ad 1. We applied the recommendations and excluded LEC from the list.

2.     The introduction should be strengthened to the scope of their investigation. By elaborating the predictors of sexual dysfunction in general population, the authors may motivate their assessment of somatic diagnoses and/or medication (including psychotropic). Nevertheless, this part could be shortened since it heavily distracts from the scope of the study. 

Ad 2. We shortened the Introduction Section trying to stick to the scope of the research.

3.     The authors declare that sexual dysfunction is predicted by overall PTSD symptom severity and by the severity of D symptoms as main hypothesis. However, this main hypothesis (in particular regarding cluster D symptoms) is not motivated in the introduction section and should be elaborated.

Ad 3. Our hypothesis regarding D symptoms is based on previous findings saying that numbing symptoms are significant predictors of SD. Numbing symptoms were categorized as C cluster symptoms in previous classification system (DSM-IV) bat categorized as a part of cluster D symptoms in DSM-5. Differences between two classification systems in relation to SD is elaborated in Discussion section. 

4.     Similar, the participants and procedure paragraph should be shortened and strengthened, e.g. the information that CHC Rijeka is providing medical care for 600000 inhabitants does not add any relevant information to the scope of the study. The information in lines 154 to 163 are also presented in table 1. Further, the authors should motivate the inclusion of the LEC-5? What does this measure contribute to the scope of the study since a diagnosis of PTSD implicates a traumatic event and this measure is not mention further?

Ad 4. We shortened the Participants Section and excluded LEC-5 from the Measures Section.

5.     What does the authors mean by ‘preliminary validation work is sufficient to make cut-point score of 33 which is chosen for the purpose of this study’ and then, restrict their statement by mentioning the variety of cut-off scores? This is confusing and the authors should instead provide a reference for the cut-point score. 

Ad 5. Corrected and reference included.

6.     In my opinion, there is no need to introduce the ICD-10 classification. This paragraph also does not add anything to the manuscript since somatic diagnoses were solely assessed by self-reports. 

Ad 6. We excluded ICD-10classification from the Measures Section.

7.     I have concerns regarding the statistical inference considering the several statistical tests / correlations conducted. Could the authors please make a statement on multiple comparison procedures?

Ad 7.  A statement in the Limitation section has been made to address the issue of multiple comparison procedures.

8.     The presentation of the results is someway confusing due to the amount of variables in the tables. The authors should consider to present solely relevant / significant results and provide the other results in the supplementary material section.

Ad 8. We extracted less relevant results and provided them in the Supplementary Material Section. Tables regarding regression analysis with significant predictors are presented in the main text as to allow for an easier overview of main results.

9 In sum, apart from statistical inference, I am struggling with the manuscript since many information are included that distract from the main results.

Ad 9. We appreciate all your recommendations and suggestions and hope that manuscript is now more focused to the main objectives and results.

Reviewer 2 Report

Thank you for the opportunity to review jcm-457992. I think the current study has laudable aim to investigate the association between Sexual Dysfunctions and PTSD. The manuscript is well written (for the most part), but we strongly recommend the service of a professional proof reading service. The Introduction adequately described the relevant preconditions of the study. Some parts of the discussion could explore in greater depth possible etiologies of the association of PTSD and SD.

Line 20 – please explain the measured constructs and then name the questionnaire in brackets, e.g. erectile dysfunction (IIEF)

Line 20 – please state that they are self-report questionnaires and interviews

Line 22 – have instead of has, just one of many mistakes. We recommend a professional proof reading service.

Line 23 – please explain cluster D

Line 38 – please refer only to a selection of most important studies or review articles

Lines 59-62 – please split the sentence.

Lines 64-65 – this sentence should belong at the beginning of the paragraph and then you should discuss the conflicting results.

Lines 72 -74 – please give an example of a possible etiology of a non-organic caused SD.

Lines 74 – 77 – please be a bit more specific, which organic condition is connected to which SD

Line 136-137 – why not eligible?

Table 1 – It would be interesting to discuss the possible confounding variable of relationship status, which is significantly different between groups, in greater depth.

Line 550 – please name a reference in which the DSM-IV diagnoses was criticized.

Line 560 – please explore in greater depth why being in a relationship may be associated to SD

Line 578 – please name a reference for the association of orgasmic functioning and antidepressant medication, a meta-analysis if possible for interested readers

Lines 616-624 – there are existing meta-analyses on the relationship of AUD and SD. Please include some references for interested readers.

Lines 633-638 – this part not so well written. Of course, clinical interviews (by well-trained personnel, who are trained about social stigmatization) are necessary to sufficiently diagnose SD. Self-report questionnaires are not a valid and reliable instrument to diagnose SD.

Line 640 – this sentence should be placed more in the beginning of this paragraph

Line 683 – I would include a preference for psychotherapeutical treatments for SD because as they have been shown to be more effective for some types of SD, have greater compliance rates and less iatrogenic effects.

Author Response

Dear Reviewer 2,

We appreciate the opportunity to resubmit our article. We described how we have responded, point by point, to the your comments. We also indicated where changes were made in the manuscript (Comments) to accommodate your requests. We provide the article with track changes and another copy with accepted changes.

Best regards,

Point by Point Reply to Review Report (Reviewer 2)

1.     Thank you for the opportunity to review jcm-457992. I think the current study has laudable aim to investigate the association between Sexual Dysfunctions and PTSD. The manuscript is well written (for the most part), but we strongly recommend the service of a professional proof reading service. The Introduction adequately described the relevant preconditions of the study. Some parts of the discussion could explore in greater depth possible etiologies of the association of PTSD and SD.

Ad 1. We accept the suggestion to conduct a professional proof reading in order to provide Improved English language and style. We also accept all other sugesstions of the reviewer explained point by point in the text below.

2.     Line 20 – please explain the measured constructs and then name the questionnaire in brackets, e.g. erectile dysfunction (IIEF)

3.     Line 20 – please state that they are self-report questionnaires and interviews

Ad 2. and 3. We applied the recommendations. We excluded LEC reffering to the comments of the Reviewer 1.  

4.     Line 22 – have instead of has, just one of many mistakes. We recommend a professional proof reading service.

Ad 4. We've made the corrections and conducted professional proof reading of the text (track changes)

5.     Line 23 – please explain cluster D

Ad 5.  We explained the cluster.

6.     Line 38 – please refer only to a selection of most important studies or review articles

Ad 6. We included only the most improtant studies.

7.     Lines 59-62 – please split the sentence.

Ad 7. We rearranged the sentence in order to try to clarify the point of the statement.

8.     Lines 64-65 – this sentence should belong at the beginning of the paragraph and then you should discuss the conflicting results.

Ad 8.  We moved the sentence to the beginning of the paragraph.

9.     Lines 72 -74 – please give an example of a possible etiology of a non-organic caused SD.

10.  Lines 74 – 77 – please be a bit more specific, which organic condition is connected to which SD

Ad 9. And 10. Examples are added

11.  Line 136-137 – why not eligible?

Ad 11. The reason for non-eligibility is added.

12.  Table 1 – It would be interesting to discuss the possible confounding variable of relationship status, which is significantly different between groups, in greater depth.

Ad 12. Added and explained under Ad 14.

13.  Line 550 – please name a reference in which the DSM-IV diagnoses was criticized.

Ad 13. Armour C, Műllerová J, Elhai JD. A systematic literature review of PTSD's latent structure in the Diagnostic and Statistical Manual of Mental Disorders: DSM-IV to DSM-5. Clin Psychol Rev. 2016 Mar;44:60-74. https://doi.org/10.1016/j.cpr.2015.12.003

14.  Line 560 – please explore in greater depth why being in a relationship may be associated to SD

Ad 14. We corrected the text for a better clarification of the result. Positive values for erectile function, orgasmic function, sexual desire, intercourse satisfaction and overall satisfaction imply better sexual functioning. Positive values for premature ejaculation imply worse sexual functioning. Along with comparison with previous research we added several sentences in lines 618-647.

15.  Lines 616-624 – there are existing meta-analyses on the relationship of AUD and SD. Please include some references for interested readers.

Ad 15. The references regarding the relationship of AUD and SD are included in lines 623-627. We included the additional reference regarding the meta-analysis.

16.  Lines 633-638 – this part not so well written. Of course, clinical interviews (by well-trained personnel, who are trained about social stigmatization) are necessary to sufficiently diagnose SD. Self-report questionnaires are not a valid and reliable instrument to diagnose SD.

Ad 16. We pointed out in the Limitations section that the findings are based on self-reported symptoms from questionnaire measures implying that sexual activity and satisfaction may be under- or overreported. We tried to emphasize the importance of the clinical interviews although there are no studies that included interviews in assessment of SD in PTSD patients (lines 713-720) We tried to be cautious with the terms regarding the diagnosis using for example provisional diagnosis of SD (Methodology and Results section) or estimation of the prevalence of SD.

17.  Line 640 – this sentence should be placed more in the beginning of this paragraph

Ad 17.  We placed the sentence according to the suggestion.

18.  Line 683 – I would include a preference for psychotherapeutical treatments for SD because as they have been shown to be more effective for some types of SD, have greater compliance rates and less iatrogenic effects.

Ad 18 We emphasized the importance of the psychotherapeutical treatments (lines 766-771 and also 761-766).

Round  2

Reviewer 1 Report

The authors could considerably improve the manuscript and handled all comments raised. 

I would recommend publication after the following minor revisions:

line 68: Please insert reference “(…) may impede sexual functioning in veterans with PTSD”

line 86: “there is an” instead of “there exists”

lines 96-99: Please rephrase this sentence

lines 173-199: Please shorten

table 1: explain the reason for providing probability for medium income in bold.

line 440: consider spaces and punctuation marks

Author Response

Dear Reviewer 1,

We appreciate your effort and you thank for your work.

1.      line 68: Please insert reference “(…) may impede sexual functioning in veterans with PTSD”

Ad 1. The sentence in line 68 does not require a reference. The reference bracket following the sentence was an editing mistake which is now corrected.

2.      line 86: “there is an” instead of “there exists”

Ad 2. We applied the recommendation.

3.      lines 96-99: Please rephrase this sentence

Ad 3. The sentence is now rephrased.

4.    lines 173-199: Please shorten

Ad. 4. The section has been shortened as suggested.

5.      table 1: explain the reason for providing probability for medium income in bold.

Ad 5. The probability for medium income was in bold unintentionally. This is now corrected, and it is in normal font as are other probabilities.

6.      line 440: consider spaces and punctuation marks

Ad 6. The sentence is corrected as suggested.

Reviewer 2 Report

I am satisfied with the changes the authors have made

Author Response

Dear Reviewer 2,

We appreciate your effort and you thank for your work.

Best regards,

Marina Letica Crepulja and colleagues.